rsob.royalsocietypublishing.org

**Subject Area:**
biophysics/structural biology

cardiac actin, ventricular myosin force–velocity, strain-sensitive regulation, actin binding myosin essential light chain, ELC N-terminus/cardiac actin contact

**Author for correspondence:**
Thomas P. Burghardt
e-mail: burghardt@mayo.edu

# Cardiac and skeletal actin substrates uniquely tune cardiac myosin strain-dependent mechanics

Yihua Wang[1], Katalin Ajtai[1] and Thomas P. Burghardt[1,2]

[1]Department of Biochemistry and Molecular Biology, and [2]Department of Physiology and Biomedical Engineering, Mayo Clinic Rochester, Rochester, MN 55905, USA

 TPB, 0000-0003-3119-4074

Cardiac ventricular myosin (βmys) translates actin by transducing ATP free energy into mechanical work during muscle contraction. Unitary βmys translation of actin is the step-size. *In vitro* and *in vivo* βmys regulates contractile force and velocity autonomously by remixing three different step-sizes with adaptive stepping frequencies. Cardiac and skeletal actin isoforms have a specific 1:4 stoichiometry in normal adult human ventriculum. Human adults with inheritable hypertrophic cardiomyopathy (HCM) upregulate skeletal actin in ventriculum probably compensating the diseased muscle's inability to meet demand by adjusting βmys force–velocity characteristics. βmys force–velocity characteristics were compared for skeletal versus cardiac actin substrates using ensemble *in vitro* motility and single myosin assays. Two competing myosin strain-sensitive mechanisms regulate step-size choices dividing single βmys mechanics into low- and high-force regimes. The actin isoforms alter myosin strain-sensitive regulation such that onset of the high-force regime, where a short step-size is a large or major contributor, is offset to higher loads probably by the unique cardiac essential light chain (ELC) N-terminus/cardiac actin contact at Glu6/Ser358. It modifies βmys force–velocity by stabilizing the ELC N-terminus/cardiac actin association. Uneven onset of the high-force regime for skeletal versus cardiac actin modulates force–velocity characteristics as skeletal/cardiac actin fractional content increases in diseased muscle.

## 1. Introduction

Cardiac myosin has a 140 kDa N-terminal globular head called subfragment 1 (S1) and an extended α-helical tail domain. Tail domains form dimers that self-assemble into myosin thick filaments with S1s projecting outward from the core in a helical array [1]. Thick filaments interdigitate with actin thin filaments in striated muscle and slide relatively during contraction [2]. S1 has the ATP and actin-binding sites (the motor) and a lever arm whose rotary movement cyclically applies tension to move a load when myosin is strongly actin bound [3]. Actin binding sites in the S1 heavy chain cover solvent-exposed regions of the molecule including structured loops (C-loop [4,5], myopathy loop and loop 3), unstructured loop 2 [6], and other peptides in the motor and within the 50 and 20 kDa tryptic fragments of the S1 heavy chain [7–9].

The lever arm converts torque generated by the motor into linear displacement (step-size) and undergoes shear strain due to the resisting stress. Strain affects the lever arm and the lever arm bound essential and regulatory light chains (ELC and RLC). The approximately 20 kDa RLC stabilizes the lever arm [10–12] and disease implicated RLC mutants lower velocity, force and strain sensitivity [13] suggesting they alter lever arm processing of shear stress [14,15]. The approximately 25 kDa skeletal myosin ELC (A1) has an N-terminus extension containing sites for ELC/actin binding localized in the N-terminal fragment

and proposed to involve actin sites near to residue D363 [16]. Subsequent work on atrial ELC localized the ELC/actin binding site to the N-terminal 11 residues [17]. A structural model for S1 bound F-actin built from skeletal S1 and skeletal actin crystal structures indicated a close match for lysine residues K3 and K4 on the ELC A1 with actin's E361 and E364 [18]. This ELC/actin interaction also has residue V6 on the ELC A1 and T358 on skeletal actin proximal and with side chain Cγ atoms separated by approximately 3.9 Å.

Cardiac and skeletal actins have greater than 98% sequence identity, differing only at positions 2 and 3 where Asp-Glu reverses to Glu-Asp and at Met299Leu and Ser358Thr for cardiac/skeletal actins. Various actin isoforms share the same conformation [19], likely ensuring near identical three-dimensional structures for the cardiac and skeletal actins. Our S1 bound F-actin homology model using cardiac ELC and cardiac actin sequences, and based on the skeletal sequences model [18], places the S358T modification in a hydrogen bond with E6 in the cardiac ELC. We find substantial differences in βmys ensemble force–velocity characteristics due to the different actin substrates that are linked to a molecular mechanism involving the S358/E6 interaction specific to cardiac muscle by using single myosin mechanical characteristics measured with the Qdot assay.

Qdot labelled actin *in vitro* motility assay (Qdot assay) has myosin immobilized on a planar glass substrate impelling single Qdot labelled F-actin with nanometre scale actin displacement, measured using super-resolution [20,21] and total internal reflection fluorescence (TIRF) microscopy [22]. The assay has negligible compliance faithfully characterizing single myosin motor displacement of actin [23–25]. We found earlier, using the unloaded Qdot assay, that ventricular cardiac myosin ELC N-terminal extension binds skeletal actin to modulate myosin step-size [23].

In experiments described here, we use the Qdot assay to compare single βmys step-size and step-frequency characteristics when moving loaded cardiac and skeletal actin. New data classification constraints better separate signal from background, giving higher signal-to-noise data from which we can resolve step-size probability densities in the sub-nanometre range. We find, for both skeletal and cardiac actin substrates, *in vitro* evidence for a 4 nm step-size pathway that is occupied only under loaded conditions and is proposed to incorporate a clutch mechanism controlling myosin slip under load to facilitate power generation under high resisting force. Our working model has myosin using one of several distinct step-size pathways from a network where an autonomous myosin molecule regulates pathway choice at two points with strain-inhibited ADP release and a strain sensing ELC N-terminus ratchet inhibiting actin detachment [25,26]. The lever arm mediates both strain sensing mechanisms.

Two actin isoforms, one dominant in skeletal the other in cardiac muscle, have a specific skeletal/cardiac actin stoichiometry of approximately 1 : 4 in a normal adult human heart [27]. It is likely that an increasing skeletal/cardiac actin stoichiometry is the programmed response of a disease-compromised heart to an inability to meet demand [28]. Earlier work suggested the α-actin isoforms have nucleotide-dependent conformational differences impacting filament stability under pathological conditions [29]. We demonstrate here significant myosin step-size distribution differences for skeletal versus cardiac actin substrate because they differently tune βmys strain-dependent mechanics. We propose that the

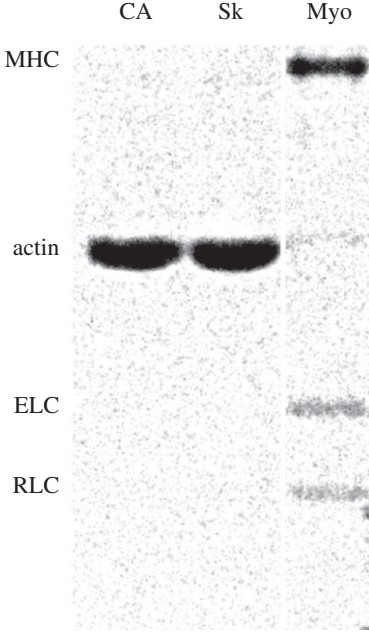

**Figure 1.** SDS-PAGE of cardiac (CA) actin, skeletal actin (Sk) and βmys (Myo). The column for βmys has regulatory and essential myosin light chains (RLC and ELC) bands indicated, and contaminating actin. Baseline subtracted absorption integrated over the actin band is approximately 10% that of the myosin heavy chain (MHC).

cardiac actin impact on cardiac myosin load sensitivity is from the unique and specific actin/ELC N-terminus contact at S358/E6. It alters cardiac motor force–velocity characteristics detected *in vitro* by both ensemble myosin motility and single myosin mechanics.

# 2. Methods

## 2.1. Protein preparation

Bovine ventricular actin and rabbit skeletal actin were prepared and purified as previously described [30]. The final preparation was stored immediately under argon gas in small aliquots in liquid nitrogen. The two actin isoforms are 99.9% pure and migrate identically in sodium dodecyl sulfate-polyacrylamide gel electrophoresis (SDS-PAGE, figure 1). Protein concentrations were determined by UV absorbance using A(1% at 290 nm) = 6.4 for actin (42 kDa). Mass spectrometry on the tissue purified bovine cardiac ventriculum actin registers 22% skeletal and 78% cardiac actin [31], in excellent agreement with previous estimates using peptide sequencing of actin from the same tissue [27]. We refer to actin purified from ventriculum as cardiac actin.

Frozen G-actin was thawed and spun at $250\,000 \times g$ for 90 min to remove denatured actin before use. Rhodamine labelling of actin filaments was performed with rhodamine–phalloidin and actin in a 1.2 : 1 molar ratio in a buffer containing 25 mM KCl, 0.2 mM CaCl$_2$, 4 mM MgCl$_2$, 10 mM DTT, 0.1 mM PMSF and 25 mM imidazole at pH 7.4 on ice for 12 h, followed by storage on ice until use within 5 days. Qdot labelling was performed as described previously [32]. Briefly, the actin filaments used for Qdot labelling were labelled with the mixture of biotin-xx phalloidin and rhodamine–phalloidin in a molar ratio of 1 : 9 and stored on ice. The overall molar ratio of phalloidin to actin and the buffer conditions were identical to those used for the rhodamine labelling.

rsob.royalsocietypublishing.org   *Open Biol.* **8**: 180143

Porcine βmys was prepared from porcine heart ventriculum as described [4,32]. Briefly, βmys was extracted from minced, washed ventriculum for 10 min at 4°C with 'Guba-Straub' solution (0.3 M KCl, 5 mM ATP, 1 mM DTT, 5 mM MgCl$_2$, 1 mM EGTA, 0.001 mg ml$^{-1}$ leupeptin in 50 mM potassium phosphate buffer, pH 6.5). Solubilized myosin was separated from the tissue by centrifugation then three precipitation cycles eliminated contaminating soluble proteins. The pellet was then dissolved in a high salt buffer (0.6 M KCl, 5 mM ATP, 1 mM DTT, 5 mM MgCl$_2$, 1 mM EGTA, 0.001 mg ml$^{-1}$ leupeptin in 50 mM Tris-HCl pH 8.0 at 4°C) followed by ultra-centrifugation (250 000×$g$ for 2 h). The upper 66% of the supernatant was collected and dialyzed overnight in storage buffer (0.6 M KCl, 2 mM DTT, 0.001 mg ml$^{-1}$ leupeptin in 25 mM Tris-HCl pH 7.4 at 4°C) followed by ultracentrifugation (250 000×$g$ for 3 h) to remove remaining actin or actomyosin impurities. The βmys was stored in sealed tubes at −20°C in 50% glycerol (vol/vol). Purified βmys has approximately 10% actin impurity judged from analysing the SYPRO Ruby stained SDS/PAGE gel shown in figure 1. Urea gel electrophoresis and mass spectrometry analysis of the βmys did not detect phosphorylated RLC in the βmys used in experiments [33].

## 2.2. Actin-activated myosin ATPase

βmys stored in 50% glycerol was precipitated with addition of 12 volumes of ice-cold water containing 2 mM DTT, collected by centrifugation, and then resuspended in 300 mM KCl, 25 mM imidazole (pH 7.4), 4 mM MgCl$_2$, 1 mM EGTA, 10 mM DTT and 0.01 mg ml$^{-1}$ leupeptin. Myosin at a final concentration of 0.6 μM was titrated with 1, 3, 5, 9, 22 and 50 μM F-actin. The ATPase assay buffer contained 25 mM imidazole (pH 7.4), 4 mM MgCl$_2$, 1 mM EGTA, 10 mM DTT, 0.01 mg ml$^{-1}$ leupeptin and a final KCl concentration of 25 mM. ATPase reaction was initiated by addition of 3 mM ATP, and the mixture was incubated at 21°C for 5 min. Inorganic phosphate measurements were performed using the Fiske and Subbarow method [34].

Actin-activated ATPase results were characterized using Michaelis–Menten kinetics,

$$V = V_{\max} \frac{[A]}{K_m + [A]},\tag{2.1}$$

for actin concentration $[A]$, $V$ the ATPase rate equal to $V_{\max}$ at saturating actin and actin-binding constant $K_m$. $V$ versus $[A]$ curves were fitted using a nonlinear algorithm to determine constants $V_{\max}$ and $K_m$.

## 2.3. *In vitro* motility and Qdot assays

*In vitro* motility and Qdot assays were performed in a flow cell using total internal reflection fluorescence (TIRF) microscopy exactly as described [23]. Motility buffer included 25 mM KCl, 25 mM imidazole (pH 7.4), 4 mM MgCl$_2$, 1 mM EGTA, 20 mM DTT, 0.01 mg ml$^{-1}$ leupeptin, 0.7% methylcellulose, 2 mM ATP, 3 mg ml$^{-1}$ glucose, 0.018 mg ml$^{-1}$ catalase and 0.1 mg ml$^{-1}$ glucose oxidase. The flow cell was infused at the start with 0.16 μM myosin where unloaded motility velocity is a maximum for both cardiac and skeletal actin. Actin sliding velocities for the *in vitro* motility assay, $s_m$, and the length of actin filaments were quantitated using FIESTA software [35].

Frictional loading assays were performed like the unloaded assays except that the flow cell was infused at the start with the mixture of myosin and α-actinin in concentrations of 0.16 μM myosin and 0–7 μg ml$^{-1}$ α-actinin (Cytoskeleton, Denver, CO).

In the Qdot assay, images were acquired with an EMCCD camera (Andor, Belfast, UK) in 45 ms intervals indicated by $\Delta t$ and using Andor's SOLIS software. Each movie recorded images for 36 s. Intensity values were converted to photons using the conversion formula in SOLIS and the images output in TIFF format for reading into IMAGEJ. We tracked movement of the Qdot labelled actin at super-resolution using the IMAGEJ plugin QuickPALM [36]. Baseline histograms corresponding to thermal/mechanical fluctuations were recorded likewise from Qdot labelled actin immobilized on the surface by myosin in the absence of ATP. All *in vitro* motility and Qdot assay experiments were conducted at room temperature (20–22°C).

## 2.4. Force calibration in the loaded actin *in vitro* motility and Qdot assays

Loaded *in vitro* motility and Qdot assay data were fitted to a viscoelastic model of frictional loads developed by Greenberg & Moore [37] and as described previously [25]. Average sliding filament motility velocity, $s_m$,

$$s_m = \frac{s_{m,\max} \cdot F_d}{F_d + s_{m,\max} c([\alpha])}\tag{2.2}$$

has $s_{m,\max}$ the velocity at zero load, $F_d$ the ensemble driving force and friction coefficient $c([\alpha])$,

$$c([\alpha]) = \frac{\kappa \cdot \xi^{5/2} \cdot \Lambda \cdot r \cdot k_A \cdot [\alpha]^{5/2}}{k_D \cdot (k_A \cdot \xi^{3/2} \cdot [\alpha]^{3/2} + k_D \cdot N_A \cdot 10^3)}$$
$$= \frac{[\alpha]^{5/2}}{c_0 [\alpha]^{3/2} + c_1}\tag{2.3}$$

for $\kappa$ system compliance associated with α-actinin (1.7 pN nm$^{-1}$), $\xi$ a constant defining the surface geometry of α-actinin on the flow cell surface (3.97 × 10$^{21}$ M$^{-1}$ m$^{-2}$), $r$ the reach of α-actinin to bind to the actin filament (82 nm), $k_A$ the second-order rate constant for α-actinin attachment to the actin filament (4 × 10$^6$ M$^{-1}$ s$^{-1}$), $[\alpha]$ molar concentration of α-actinin and $k_D$ α-actinin detachment rate (9.6 s$^{-1}$). The constants are suggestions based on the literature and identical to those described [37]. Other known quantities in equation (2.3) are Avogadro's number, $N_A$, and actin filament length in our system, $\Lambda \approx 1$ μm. Coefficients $c_0$ and $c_1$ in equation (2.3) are optimized when fitting data (discussed below) thus providing an experimental value for their relationships to the estimated constants (i.e. system compliance, surface geometry, ...) mentioned above and as implied by equation (2.3).

Frictional force, $F_f$, exerted by the α-actinin in the loaded motility assay,

$$F_f = s_m c([\alpha])\tag{2.4}$$

relates force and velocity in the Hill equation [38] for normalized force and velocity,

$$\frac{s_m}{s_{m,\max}} = \frac{K(1 - F_f/F_0)}{(K + F_f/F_0)}\tag{2.5}$$

and implies power output, $P$,

$$P = F_f s_m = F_f \frac{K(1 - F_f/F_0)}{(K + F_f/F_0)} s_{m,\max},\tag{2.6}$$

rsob.royalsocietypublishing.org   Open Biol. **8**: 180143

rsob.royalsocietypublishing.org    Open Biol. **8**: 180143

for $F_0$ isometric force and dimensionless constant $K$ proportional to the myosin attachment rate $f_{APP}$ [39]. Assuming $F_d = F_0$, we estimate free parameters $F_0$, $K$, $c_0$ and $c_1$ using linear programming with constraints from the raw data consisting of $[\alpha]$ versus $s_m$ data points for skeletal and cardiac actin, equations (2.2)–(2,5), and requiring that all fitted parameters are greater than or equal to 0. Fitted constants $F_0$ and $K$ have independent values for skeletal or cardiac actin while $c_0$ and $c_1$ are identical for both species. In addition, the maximum unitary force generated by βmys is likely to be approximately 2 pN [40] and the number of βmys simultaneously impelling a 1 μm long (skeletal) F-actin is approximately 15 [33], implying an upper limit constraint on $F_0$ of 60 pN (twice the estimated maximum of 30 pN). When solved, parameters calibrate the frictional force (in pN) using equations (2.3) and (2.4) and give best estimate fitted curves for $s_m$ versus $[\alpha]$ (equations (2.2) and (2.3)), $s_m$ versus $F_f$ (equation (2.5)) and $P$ versus $F_f$ (equation (2.6)).

## 2.5. Qdot assay event–velocity histogram simulation

Quantitation of Qdot event–velocity histograms was performed as already described [25] except when estimating baseline thermal/mechanical fluctuation amplitudes. A new method provided better estimates as described next.

Experimental data, $\mathbf{v}_{obs}$ and $\mathbf{v}_{t/m}$, and simulated data, $\mathbf{v}_{sim}$, are event–velocity histograms in a vector representation. The $\mathbf{v}_{obs}$ is the observed velocities of Qdot labelled actin in the presence of ATP. The $\mathbf{v}_{t/m}$ velocities quantitate actin movement from thermal/mechanical fluctuations measured independently from Qdot labelled actin immobilized on the surface by myosin in the absence of ATP. Simulated velocities, $\mathbf{v}_{sim}$, quantitate actin movement by active myosins exactly as described [25]. These event–velocity histograms are linearly related by

$$\mathbf{v}_{obs} = c_2\mathbf{v}_{sim} + c_3\mathbf{v}_{t/m}, \tag{2.7}$$

for $\mathbf{v}_{sim}$ and $\mathbf{v}_{t/m}$ $\ell_1$ normalized vectors, and, $c_2$ and $c_3$ unknown scalar constants equal to the total myosin-based and thermal/mechanical-based actin displacement events. Unknowns $c_2$ and $c_3$ are solved optimally using linear programming under the equality constraint that known total observed events = $c_2 + c_3$ and the inequality constraints $c_2 \geq 0$ and $c_3 \geq 0$.

## 2.6. Four-pathway contraction model

Earlier work using the unloaded *in vitro* Qdot assay on βmys [32] combined with the *in vivo* single cardiac myosin imaging of zebrafish embryo hearts [26] suggested a four-pathway contraction model like that in figure 2 for generating 8 (blue pathway), 5 (green), 5 + 3 (green-yellow) and solo-3$^+$ (red) nm myosin step-sizes [25]. The contraction model shown here is modestly changed from the earlier work mainly in the solo-3$^+$ nm pathway where the slip transition is defined in more detail due to new data as described below. We summarize the four-pathway contraction model (figure 2) briefly below and will refer to it throughout to clarify subsequent discussion of cardiac myosin's ability to step with multiple unitary step-sizes.

In figure 2, weak actin binding myosin reversibility (indicated with ↔) preemptively avoids pathways that do not complete cycles under increasing load. Under loaded conditions, flux through the cycle is checked at two strain-inhibited points indicated by the lightning bolts. The traditional lever arm strain checkpoint inhibits ADP release and hence ATP dissociation of actomyosin following the 5 nm step-size (green path) sometimes sending flux towards the 3 nm step-size (yellow branch). The ELC-ratchet strain checkpoint regulates the solo-3$^+$, 8 and 5 + 3 nm pathways (red, blue and green-yellow). The latter senses tension in the actin-bound ELC N-terminus such that the slack ELC extension (blue, when muscle is unloaded or rapidly shortening under low load) allows ATP binding and quick detachment from actin to complete the cycle, the moderately tense ELC extension (green, when muscle is under moderate loads approaching isometric) partially inhibits ATP binding and detaches more slowly from actin to exert static myosin tension contributing to force on slowly translating muscle filaments, and the maximally tense ELC extension (red, when muscle is near-isometric) inhibits ATP binding and detaches slowest from actin to exert static myosin tension contributing more of the force on the static muscle filaments. The ELC-ratchet tends to filter out longer step-sizes as tension rises since they will transfer a larger tension to it. The lever arm mediates both ADP release strain inhibition and ELC-ratchet strain inhibition.

Strongly bound myosin motors contributing static tension are the force bearing 0 length step-size myosins not explicitly depicted in figure 2 (see rather fig. 8 in [26]). Modulating flux through two strain-dependent steps with different inhibitions adjusts average step-size. The favoured pathway at high tension (greater than 80% of cycles) involves the solo-3$^+$ nm step via the red pathway. It releases Pi with slight net forward movement probably by slipping at high tension but then completes a 3 nm step. Results introduced here suggest 4 nm slippage of a 5 nm step extends the solo-3$^+$ nm step-size to 4 nm. Slip distances of 2–8 nm were observed in synthetic myofilaments under tension (see fig. 1$e$–$g$ in [41], larger slips were observed earlier [42]). This pathway is populated even at the highest tension implying the ELC ratchet strain checkpoint is less inhibiting than the ADP release inhibiting checkpoint in near-isometric contraction [43].

## 2.7. Homology modelling

A molecular model for the skeletal myosin S1(A1) isoform bound to skeletal F-actin was described with 14 actin monomers forming the actin filament and two S1s with light chains bound to the filament [18]. The model is based on crystal structures for skeletal myosin (2mys) [44] and skeletal actin (1atn) [45], and electron density maps for S1 bound to F-actin [46–48]. The ELC(A1) N-terminal extension is missing from the model structures and was constructed using structural pattern searches and energy minimization. Three actin monomers contacting S1 (two contacting the heavy chain and another contacting the ELC N-terminus) were extracted from that structure to use as the model for the all-cardiac isoform structure. Cardiac actomyosin was homology modelled using MODELLER [49] and human sequences ACTC1, MYH7, MYL2 (RLC) and MYL3 (ELC).

## 2.8. Statistics

Actin-activated ATPases were measured four times with at least two different actin and two different βmys preparations. Significance testing generally uses one-way or two-way ANOVA with Bonferroni or Tukey–Kramer post-tests for significance.

Actin motility velocity datasets consist of 7–31 acquisitions (one acquisition is one *in vitro* motility movie) per

rsob.royalsocietypublishing.org Open Biol. 8: 180143

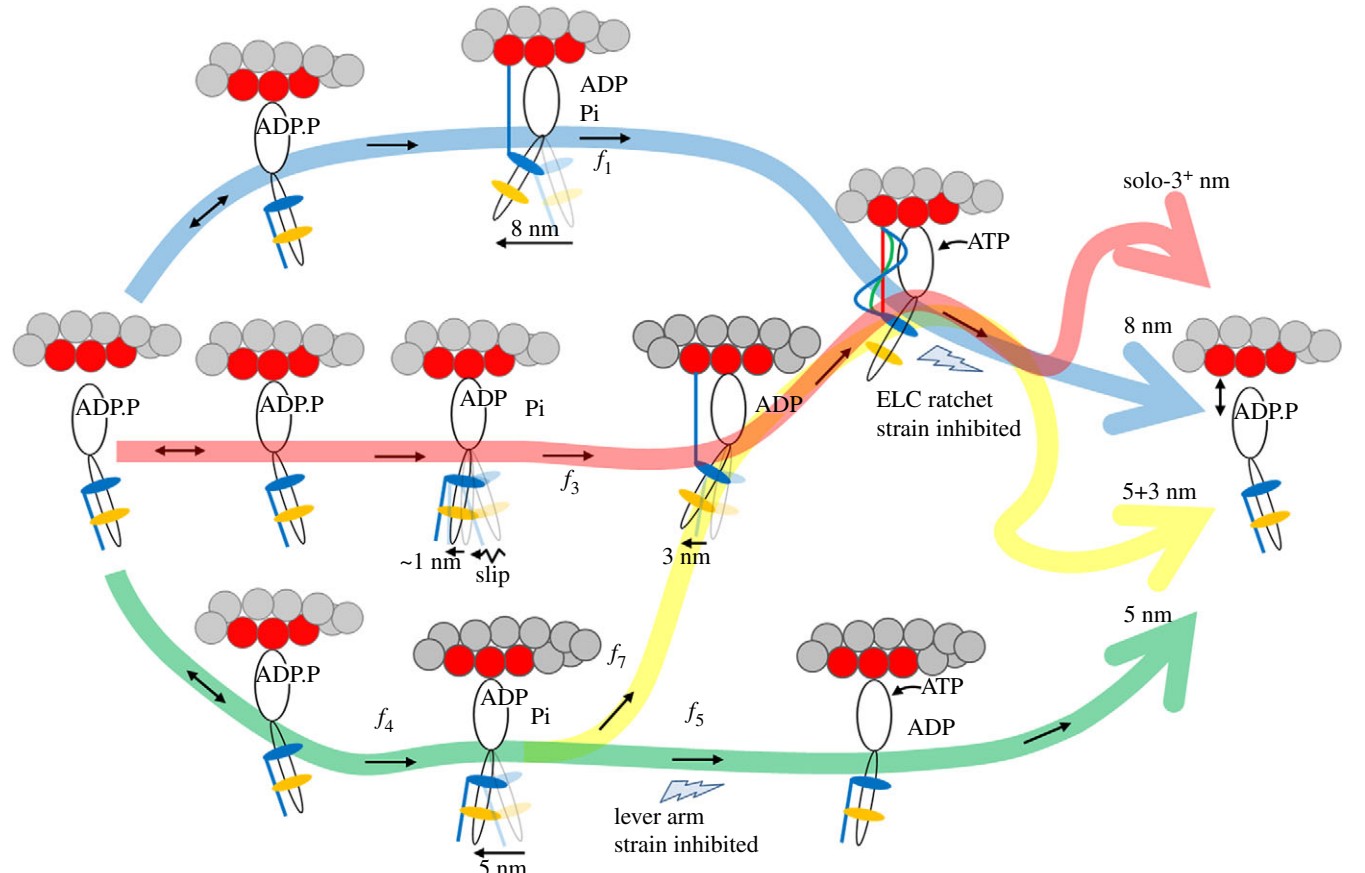

**Figure 2.** The four-pathway network for cardiac myosin unitary steps. An unloaded myosin powerstroke generally has two sequential enzymatic steps with Pi release driving the larger lever arm swing (nominal 5 nm step-size) followed by the ELC N-terminus binding actin and ADP release driving the smaller lever arm swing (3 nm step-size). The blue pathway has an 8 nm step-size where large and small steps are tightly coupled for the maximum lever arm swing. The green pathway is predominant with the 5 nm step accompanying product release. Occasionally Pi release with the 5 nm step is not immediately followed by ADP release because of lever arm strain inhibition at the lower thunderbolt allowing time for the subsequent ELC N-terminus binding and 3 nm step in the yellow pathway. Flux through the network pathways is variously distorted by loading. The red pathway denotes a solo-3$^+$ nm step-size where myosin partially slips to releases Pi with approximately 1 nm net forward movement but then the ELC N-terminus binds actin permitting ADP release and completion of a 3 nm step. Myosin in the red pathway is observed here as a new 4 nm step-size occurring only in a high-force regime defined in the text. The ELC-ratchet strain inhibiting filter at the upper thunderbolt regulates the 3 nm part of the 5 + 3 nm pathway as well as the solo-3$^+$ and 8 nm pathways (yellow, red and blue). ELC ratchet strain inhibits ATP binding and ELC N-terminus detachment from actin maintaining tension at peak isometric force. The lever arm strain inhibition at the lower thunderbolt regulates the 5 nm pathway (green) with strain inhibition of ADP release. The upper and lower thunderbolts are competing strain regulated checkpoints that modulate step-frequencies for quickly responding to changing force–velocity demands. Myosin flux through the network is denoted in the figure with $f_i$ for $i = 1, 3, 4, 5$ and 7, and identically to earlier work [25]. Flux calculations were not attempted here because the constraints limiting the flux values are sometimes qualitative as described in the text.

actin isoform and α-actinin concentration. Qdot assay datasets consist of 8–16 acquisitions per actin isoform and α-actinin concentration. Datasets for motility velocity and Qdot assays were accumulated from at least two different actin and two different βmys preparations.

We simulated data for each event–velocity histogram independently to estimate single myosin mechanical characteristics consisting of three step-sizes and three step-frequencies. We compared fitted parameters as categorical variables in factor 1 for the 8–16 independent event–velocity histograms in a dataset in factor 2 using two-way ANOVA with Bonferroni or Tukey–Kramer post-tests for significance at the 0.05 level. This test indicated no significant difference among event–velocity histograms in datasets, hence parallel acquisitions were pooled.

Simulated data ensembles were created by using best fitting event–velocity histogram simulations generated for a Qdot assay dataset, where simulations are combined linearly to approximate the event–velocity histogram from the pooled data with coefficients greater than or equal to 0 while

minimizing the $\chi^2$ goodness-of-fit test with all points equally weighted. The number of best fitting simulations combined is less than or equal to the number of independent event–velocity histograms in a dataset.

## 2.9. Effect of actin mixtures

Actin purified from rabbit skeletal muscle is the homogeneous skeletal isoform [50] while that from bovine ventriculum contains 22 and 78% skeletal and cardiac isoforms, respectively [31]. Measured characteristics using actin-activated ATPase, actin motility and Qdot labelled actin motility contrast responses of pure skeletal and heterogeneous cardiac actin samples. We show that, for the two actin species, actin-activated ATPase is statistically identical while other quantities are significantly different.

We investigated the effect of actin mixtures on motility velocity by testing sensitivity of parameters estimated from the motility using actin velocity versus loading force data

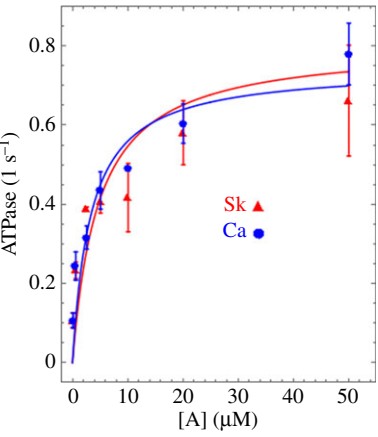

**Figure 3.** Actin-activated βmys ATPase in inverse seconds versus micromolar actin concentration [A] for cardiac (Ca, blue) and skeletal (Sk, red) actin. Error bars show standard deviation for sampling statistics given in table 1 and under experimental conditions given in Methods. Fitted curves use equation (2.1). Difference between ATPase rate versus [A] for cardiac and skeletal actin is negligible.

corrected by

$$s_c = \frac{s_o - x_s\, s_k}{1 - x_s},\qquad(2.8)$$

where $x_s$ is the fraction of skeletal actin in the mixture (approx. 0.22), $s_c$ and $s_k$ pure unknown cardiac and known skeletal actin isomer velocities versus loading force, and $s_o$ the observed actin velocity versus loading force from ventriculum purified actin. We similarly estimated the effect of the actin mixture on Qdot assay measured ensemble averaged quantities including filament velocity, step-size ($\langle d \rangle$) and duty ratio ($\langle f \rangle$) using equation (2.8) and substituting $s$ with the mentioned Qdot assay measured quantity.

Presumption of equation (2.8) has actin monomer subunits impacting velocity independently when the isoforms comingle in cardiac F-actin. We show below that it is a reasonable supposition for this system.

# 3. Results

## 3.1. Actin-activated ATPase of cardiac myosin

Figure 3 shows the actin-activated myosin ATPase versus actin concentration, [A], for skeletal (red triangles) and cardiac actin (blue circles) with error bars indicating standard deviation. Curves were fitted using equation (2.1) to obtain the Michaelis–Menten parameters $V_{max} = 0.762 \pm 0.099\ s^{-1}$, $K_m = 3.6 \pm 1.5\ \mu M$ for cardiac actin, and $V_{max} = 0.816 \pm 0.071\ s^{-1}$, $K_m = 5.29 \pm 1.89\ \mu M$ for skeletal actin and as indicated in table 1.

We tested the significance of data in figure 3 using two-way ANOVA with factor 1 proteins skeletal actin and cardiac actin, and factor 2 the actin concentration [A]. Differences between actin-activated myosin ATPases for the two actin isoforms are not significant for confidence level $p < 0.05$.

Cryo-EM data indicate rigor actomyosin contacts (in the presence of tropomyosin) between myosin loop 2 and N-terminal actin residues 1–4 [51]. The actin N-terminus contains skeletal/cardiac actin substitutions Glu2Asp and Asp3Glu while loop 2 links the 50 and 20 kDa proteolytic fragments of S1. Loop 2 substitutions dramatically impact actin-activated ATPase and actin

binding affinity [52,53], implying loop 2/actin N-terminus contacts participate integrally in normal motor function, and the potential for skeletal versus cardiac actin motor function differences. However, data in figure 3 suggest the loop 2/actin interaction is negligibly impacted by the skeletal/cardiac actin Glu2Asp and Asp3Glu substitutions.

A single cardiac myosin contacts three actins while the ELC N-terminus extension binds just one. Assuming Glu2Asp and Asp3Glu substitutions do not affect the actomyosin interaction, then each βmys has either a cardiac or skeletal actin-type contact but never a hybrid of forms. Then the average number of skeletal actomyosin contacts during motility is linear with the amount of skeletal actin and justifies the presumption of equation (2.8).

## 3.2. Loaded motility

Figure 4*a* shows βmys *in vitro* motility velocity, $s_m$, versus α-actinin concentration, [α], for cardiac (Ca, blue circles) and skeletal actin (Sk, red triangles). Figure 4*b* shows the same data in velocity versus normalized frictional force described by the Hill equation (equation (2.5)). Figure 4*c* indicates the same data in a power versus normalized frictional force representation using equation (2.6). βmys generates approximately twofold more peak power when moving skeletal compared to cardiac actin. Error bars indicate standard deviation for 7–31 acquisitions at each α-actinin concentration. We tested the significance of $s_m$ versus [α] data (*a*), $s_m$ versus $F_f/F_0$ data (*b*) and power versus $F_f/F_0$ data (*c*) using two-way ANOVA with factor 1 the skeletal or cardiac actin, and factor 2 the [α] or $F_f/F_0$ values. Data differ significantly with confidence level $p < 0.01$.

Skeletal and cardiac actin fitted curves in figure 4*a–c* are optimized with free parameters $F_{0k}$, $F_{0c}$, $K_k$, $K_c$, $c_0$ and $c_1$ constrained in equations (2.2)–(2.6), where $F_{0k}$ and $F_{0c}$ are isometric forces for myosin moving skeletal and cardiac actin, respectively. Dimensionless constants $K_k$ and $K_c$ for skeletal and cardiac actin (equation (2.5)) are proportional to myosin attachment rate $f_{APP}$. We find skeletal actin supports a significantly larger isometric force than actin purified from cardiac ventriculum. $K_k$ and $K_c$ are 10.1 and 7.0 for the skeletal and cardiac actin and are approximately 40 times larger than that reported from intact slow skeletal muscle [39,54]. Rate $f_{APP}$ distinguishes the actin isoforms suggesting the skeletal actin promotes the force generating state as also implied by power in *c*. Mean and standard deviation values for $F_{0k}$, $F_{0c}$, $K_k$ and $K_c$ are summarized in table 1. Fitted parameters $c_0$ and $c_1$ reflect on the various quantities defining the friction coefficient in equation (2.3). They suggest that the values mentioned in Methods contributing the ratios $k_D\, k_A\, \xi^{3/2}$ and $k_D^2\, N_A\, 10^3$ over $\kappa\, \xi^{5/2}\, \Lambda\, r\, k_A$ and corresponding to constants $c_0$ and $c_1$ in equation (2.3) underestimate their true value in the motility assay by approximately 100 fold.

We estimated the impact of the skeletal actin component in ventriculum purified actin using corrected cardiac actin velocity data from equation (2.8), constants $c_0$ and $c_1$ from the fitted curves in figure 4*a–c* (i.e. the implicit assumption that $k_D$, $k_A$, $\xi$, $\kappa$, $r$ and $\Lambda$ are equal for skeletal and cardiac actin) and with free parameters $F_{0k}$, $F_{0c}$, $K_k$ and $K_c$ optimized while constrained by equations (2.2)–(2.6). We find the free parameters do not differ significantly for cases using observed or corrected cardiac actin velocity data. Results for corrected cardiac actin velocity data are also indicated in

rsob.royalsocietypublishing.org    *Open Biol.* **8**: 180143

rsob.royalsocietypublishing.org Open Biol. **8**: 180143

**Table 1.** Mechanical characteristics of βmys for two actins.[a]

| | skeletal actin | cardiac actin | p< |
|---|---|---|---|
| | actin-activated myosin ATPase | | |
| $V_{max}$ (s$^{-1}$) | 0.82 ± 0.07 (N = 4) | 0.76 ± 0.10 (4) | ns @ 0.05 |
| $K_m$ (μM) | 5.2 ± 1.9 | 3.7 ± 1.5 | ns @ 0.05 |
| | peak *in vitro* motility velocity for unloaded actin | | |
| $s_{m,max}$ (μm s$^{-1}$) | 0.20 ± 0.01 (N = 31) | 0.16 ± 0.01 (23) | 0.01 |
| | isometric force ($F_0$) in pN and dimensionless attachment rate ($K$) for motility data | | |
| $F_0$ | 18.4 ± 3.7 (N = 12) | 14.3 ± 2.8 (12) | 0.01 |
| $K$ | 10.1 ± 4.2 | 7.0 ± 4.0 | 0.01 |
| $F_0$ corrected | 21.0 ± 2.3 (12) | 15.3 ± 1.5 (12) | 0.01 |
| $K$ corrected | 10.3 ± 4.0 | 3.7 ± 2.5 | 0.01 |
| | isometric force ($F_0$) in pN and dimensionless attachment rate ($K$) for Qdot assay motility data | | |
| $F_0$ | 38.5 ± 9.8 (N = 8) | 29.5 ± 10.5 (8) | 0.01 |
| $K$ | 18.0 ± 0.1 | 14.0 ± 5.1 | 0.01 |
| $F_0$ corrected | 36.5 ± 9.4 (8) | 29.1 ± 12.3 (8) | 0.01 |
| $K$ corrected | 18.0 ± 0.1 | 12.0 ± 6.1 | 0.01 |

[a]Average values and standard deviation indicated for ($N$) replicates. ANOVA significance under the $p <$ heading indicates confidence level for distinguishing skeletal from cardiac actin or ns for no significant difference. $F_0$ or $K$ corrected refers to optimization of equations (2.2) − (2.6) using corrected cardiac actin data from equation (2.8).

table 1. The skeletal actin impurity in the actin purified from ventriculum has negligible impact on *in vitro* motility.

Qdot labelled skeletal and cardiac actin velocity data in figure 4 $d−f$ parallel those in $a−c$ except that [α] sampling is limited to 0, 1, 3 and 5 μg ml$^{-1}$. Skeletal and cardiac actin fitted curves in figure 4$d−f$ were surmised by methods identical to those used to fit the data in figure 4$a−c$. We find the Qdot labelling alters slightly the α-actinin/actin interaction reflected in modest changes in the optimized values for $c_0$, and $c_1$ while other fitted parameters are practically identical but with larger standard deviation estimates. Mean and standard deviation values for $F_{0k}$, $F_{0c}$, $K_k$ and $K_c$ are summarized in table 1.

The Qdot labelled actin data were collected for the single myosin experiments; however, in their alternate form summarized by figure 4$d−f$ we use them to evaluate the impact of the skeletal actin component in ventriculum purified actin on Qdot labelled actin motility. Results compared in table 1 show that the fitted parameters pertaining to myosin function do not differ significantly for the observed or corrected (using equation (2.8)) cardiac actin velocity data. This ensemble measurement shows that isometric force and actin attachment rate in equations (2.2)–(2.6) for ventriculum purified actin are negligibly impacted by the small skeletal actin component, suggesting pooled single myosin characteristics measured with the Qdot assay are also likely to be negligibly impacted.

## 3.3. Loaded Qdot assay

The Qdot assay has Qdot labelled actin filaments approximately 1 μm long translating over surface bound βmys. Figure 5 shows the Qdot assay pooled data event–velocity histogram from 8 to 16 acquisitions (N) for βmys moving actin from cardiac (figure 5$a$) or skeletal (figure 5$b$) muscle and for increasing frictional load indicated in pN. Event–velocity histogram domains cover 0 to approximately four natural velocity units (vu), where $(d_I/\Delta t) = 1$ for $d_I$ the intermediate step-size, usually approximately 5–6 nm, and frame capture interval $\Delta t = 45$ ms. Summary data (solid squares connected by dashed line) have the baseline contribution from thermal/mechanical fluctuations already subtracted to show motility due to myosin activity. Peaks or inflection points below 2 vu are short ($d_S$ indicated with short red up arrow), intermediate ($d_I$ indicated with longer green down arrow) and long ($d_L$ indicated with longest blue up arrow) unitary step-sizes in nanometres. Some unitary step combinations are also indicated. Simulation of each event–velocity histogram acquisition is summarized for the pooled data by the solid line overlaying summary data [25].

Figure 6 shows the step-frequency expectations (lines), dimensionless relative expectation values {$\omega_S$, $\omega_I$, $\omega_L$}, and their standard deviations for N acquisitions for the short (S), intermediate (I) and long (L) unitary steps. Step-frequency expectations are derived from simulation of each event–velocity histogram acquisition [25]. The expectation curves indicate the relative probability for step-frequency along the abscissa. The area under the colour-coded curves for the S, I and L steps equals the expectation values $\omega_S$, $\omega_I$ and $\omega_L$, respectively. The sum $\omega_S + \omega_I + \omega_L = 1$ in each panel.

Figure 7$a$ shows average unitary displacement in nanometres, $\langle d \rangle = \omega_S \cdot d_S + \omega_I \cdot d_I + \omega_L \cdot d_L$ [25], and figure 7$b$ shows the duty ratio, $\langle f \rangle$ [25], for skeletal (red) and cardiac (blue) actin derived from simulation of each event–velocity histogram acquisition. Error bars show standard deviation for $N$ acquisitions.

Under unloaded conditions (top row in figures 5 and 6), unitary step-sizes indicate a significant 0.4–0.5 nm shift to longer displacements for the cardiac actin while step-frequencies are identical for the two actins. Average displacement (figure 7$a$) also reflects a longer average unitary displacement for the unloaded ($F_f = 0$) cardiac actin. The different characteristic displacement is reflected equally in each unitary step-size, implying a geometrical difference in the actomyosin complex

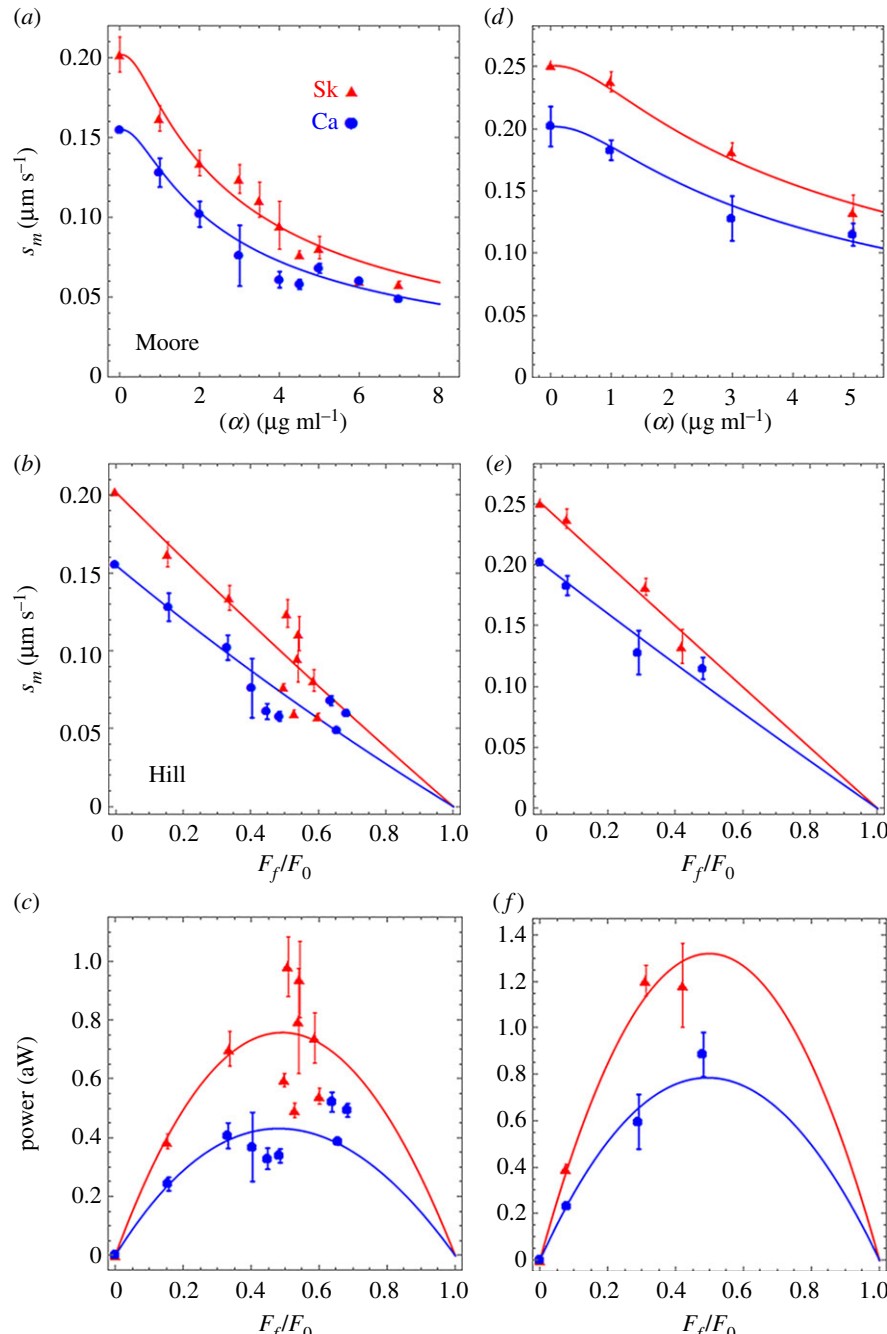

**Figure 4.** βmys loaded *in vitro* motility assay for skeletal (Sk, red) or cardiac (Ca, blue) actin. (*a*) Motility velocity, $s_m$, versus α-actinin concentration. (*b*) Motility velocity as in (*a*) versus normalized frictional force $F_f/F_0$ for $F_f$ the frictional loading force and $F_0$ isometric force. (*c*) Power versus $F_f/F_0$. Error bars show standard deviation for 7–31 acquisitions at each α-actinin concentration and under experimental conditions given in Methods. Fitted curves are based on equations (2.2)–(2.6) applied to data as described in Methods. Experimental data appear as discrete points with error bars and fitted curves as solid lines. Significance testing of cardiac versus skeletal datasets in (*a*–*c*) indicates they differ significantly in each case with confidence level $p < 0.01$. Panels (*d*–*f*) are identical to (*a*–*c*) except for Qdot labelled actin under fewer and slightly different loads and with error bars showing standard deviation for 8–16 acquisitions.

probably unrelated to individual pathways in figure 2. The overall three-dimensional structure of cardiac F-actin probably differs from skeletal F-actin giving the lever arm swing a longer projection in the direction of F-actin movement.

Although three distinct unitary step-sizes characterize the Qdot labelled actin motility in simulation, we now enjoy higher spatial resolution and observe a more complex pattern dependent on load where new step-sizes emerge and others decline, demonstrating changing flux through alternative pathways in figure 2. Below 5 pN resisting force, the cardiac actin event–velocity histogram maintains the familiar short, intermediate and long step-sizes of approximately 3, approximately 5 and approximately 8 nm attributed to 5 + 3 nm (green-yellow),

5 nm (green) and 8 nm (blue) pathways in figure 2 but not including the solo-3⁺ nm pathway (red) that contributes at higher resisting force [25]. Low-force unitary step-sizes slightly shorten as resisting force increases as expected. This is the low-force regime for cardiac actin. At 5 pN resisting force, a nascent 4 nm step-size emerges in the cardiac actin event–velocity histogram (red asterisk, figure 5). It is established and significant in the event–velocity histogram by 8 pN resisting force (figure 5) signalling onset of the high-force regime for cardiac actin. During the transition to the high-force regime, the canonical 3 nm step-size declines in favour of a new approximately 4 nm step-size. Step-size does not lengthen under resisting force but rather shortens as observed for the cardiac actin in the low-force regime and

rsob.royalsocietypublishing.org   Open Biol. **8**: 180143

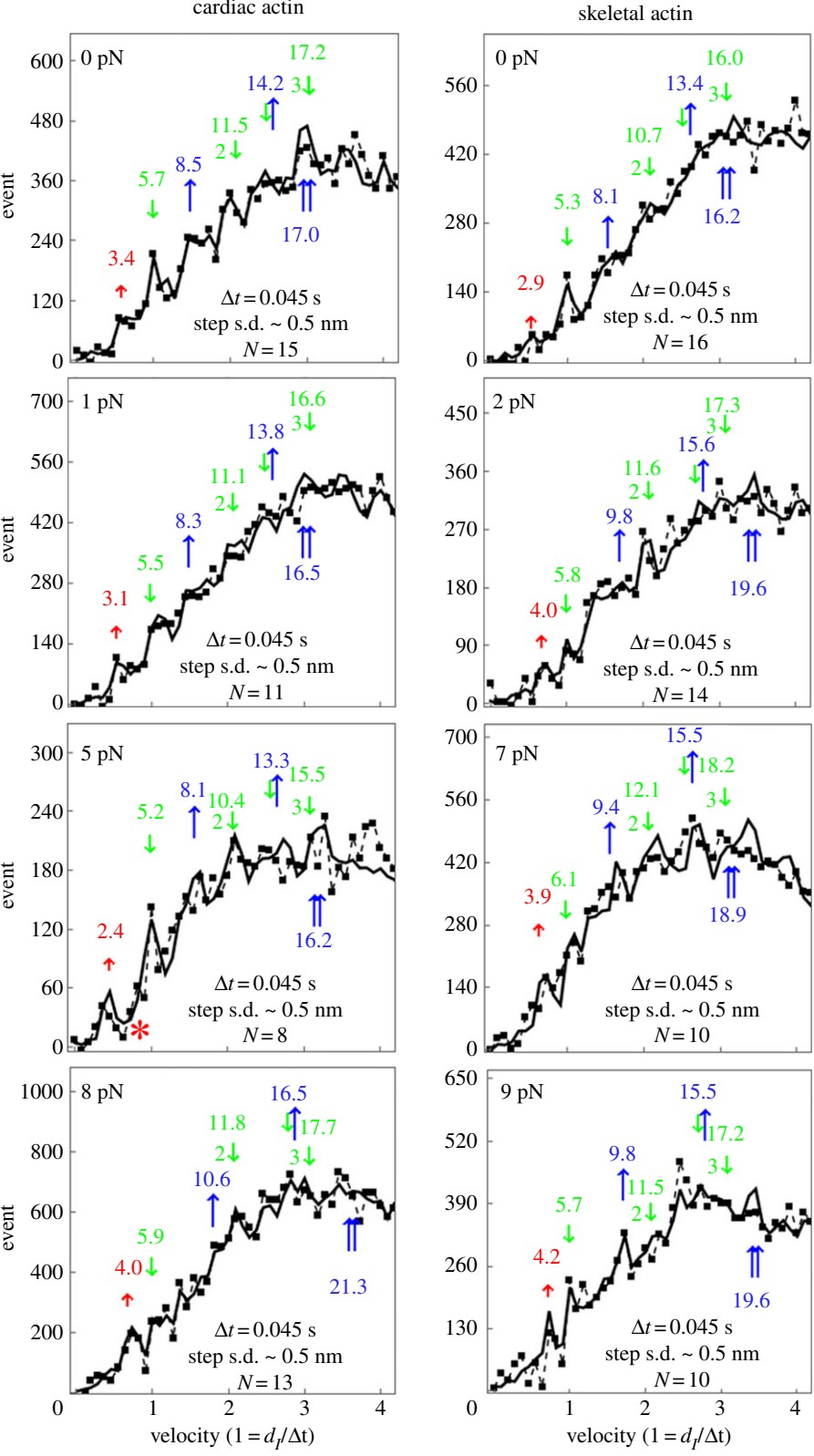

**Figure 5.** Event versus velocity histograms (solid squares connected by dashed lines) for βmys moving cardiac (*a*) or skeletal (*b*) actin and for frictional actin loading, $F_f$, from 0 to 9 pN. Black solid lines are simulations performed as described in Methods and used to estimate step-size (at arrows) and step-frequency in figure 6. Natural velocity units (vu) have 1 vu $= (d_l/\Delta t)$ for $d_l$ the intermediate step-size (green down arrow at approx. 5–6 nm) between the short (red up arrow at approx. 2–4 nm) and long (blue up arrow at approx. 8–9 nm) step-sizes. Step-sizes have a standard deviation of approximately 0.5 nm for the 8–16 replicates.

elsewhere [55], hence the new 4 nm step-size is from the solo-3$^+$ nm red pathway, a strain-shortened 5 nm step-size from the green pathway, or contributions from both (figure 2). Skeletal actin data transitions to the high-force regime at or before the lowest resisting force of 2 pN.

The solo-3$^+$ nm pathway produces a 4 nm step-size when slip distance is approximately 4 nm, well within distance

ranges previously reported [41,42]. Disappearance of the approximately 3 nm step-size (from 5 + 3 nm yellow pathway) also signals reduction in the 5 nm step-size (from 5 nm green pathway), especially since lever arm strain-inhibited ADP release would tend to boost 3 nm step-frequency. This observation is coupled with emergence of the approximately 4 nm step-size, implying flux normally flowing into the green

rsob.royalsocietypublishing.org    Open Biol. **8**: 180143

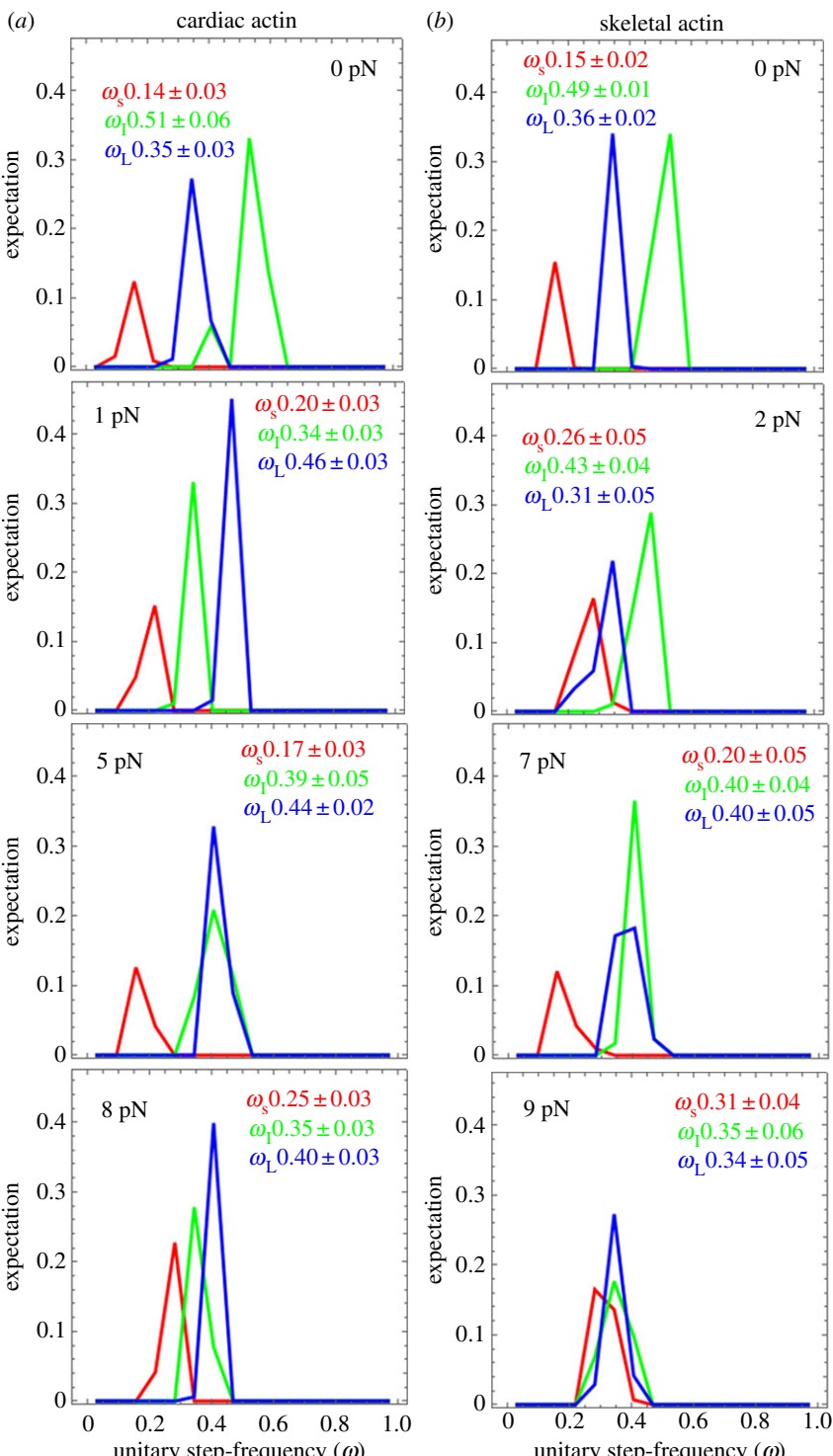

**Figure 6.** Step-frequencies for cardiac (*a*) and skeletal (*b*) actin with frictional loading indicated from 0 to 9 pN. Curves are from simulation of the corresponding event−velocity histograms in figure 5 and as described in Methods. Errors are standard deviations for the 8−16 replicates indicated for each species in figure 5. Leading step-size choice for βmys moving cardiac actin is always the long (blue) step-size. By contrast, βmys moving skeletal actin is always the intermediate (green) step-size. These selections reflect competitive regulation of the myosin mechanical characteristics by the ELC ratchet and lever arm strain mechanisms.

pathway at low force preemptively diverts to the solo-3$^+$ nm (red) pathway and suggesting the approximately 4 nm step-size has minimal or no contributions from a strain-shortened 5 nm step-size. Thus the solo-3$^+$ nm pathway solely causes emergence of the approximately 4 nm step-size at resisting forces greater than or equal to 5 pN for cardiac actin and at greater than 0 load for skeletal actin.

In the high-force regime, the intermediate step-size category has a characteristic displacement slightly rising to approximately 6 nm, suggesting residual 5 nm step-size displacements combine with strain-shortened 8 nm step-size

displacements from the blue pathway. The long step-size category shows displacement at 9–11 nm, about equal to the sum of steps from the short and intermediate categories. The latter is unlikely to be unitary but rather separate unitary events from the short and intermediate categories sometimes falling into one frame capture interval and observed as a single peak in the event−velocity histogram.

Step-frequencies in figure 6 indicate cardiac actin shifts the leading step-frequency to the long step-size with application of the lowest resisting force. This was not observed previously and unique for the cardiac actin substrate. We suggest below that it

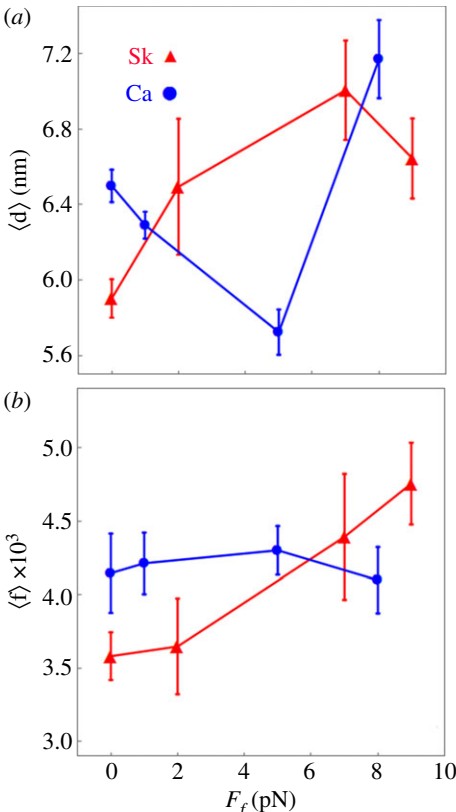

**Figure 7.** *In vitro* single βmys mean step-size ⟨d⟩ (*a*) and mean duty ratio ⟨f⟩ (*b*), for skeletal (red) and cardiac (blue) actin substrates measured with the Qdot assay under increasing load. Data points are derived from simulation of the corresponding event–velocity histograms in figure 5 and as described in Methods. Errors are standard deviation for 8–16 replicates. Two-way ANOVA tested significance for factor 1 proteins skeletal actin and cardiac actin, and factor 2 the frictional force $F_f$ shows that ⟨d⟩ does not while ⟨f⟩ does differ significantly with confidence level $p < 0.01$. Nevertheless, ⟨d⟩ appears to signal the drag force-delayed onset of the high-force regime for myosin impelling cardiac actin since ⟨d⟩ increases dramatically to its skeletal value at the highest drag force.

is because cardiac actin stabilizes the ELC N-terminus binding by an extra hydrogen bond at Ser358 that is absent for Thr358 in skeletal actin. The short step-size frequency increases with loading force except at the transition from low- to high-force regimes in cardiac actin (approx. 5 pN resisting force) where the low-force short step-size probability is phasing out in favour of the high-force short step-size probability. The three step-size simulation inadequately quantitates flux through the system pathways during transition at 5 pN (figure 5). Skeletal actin is always in the high-force regime (except at 0 load) where assigning density to unitary steps is ambiguous, as discussed above, hence skeletal actin step-frequencies in figure 6 are qualitative.

Duty ratios (⟨f⟩) in figure 7*b* compare cardiac and skeletal actin. Skeletal actin is always in the high-force regime (except at 0 load) where assigning density to unitary steps is ambiguous, as mentioned above, hence quantities assigned to skeletal actin in figure 7 are qualitative. The cardiac actin isoform has constant or decreasing ⟨f⟩ over the frictional loading forces tested while skeletal actin has increasing ⟨f⟩. The former is remarkable because it implies that the increasing load does not raise $t_{on}$ as would be expected if lever arm strain-inhibited ADP release was the sole regulatory mechanism adjusting force–velocity to match load. Instead, the system uses actively competing mechanisms of lever arm strain-inhibited ADP release and the ratcheting ELC N-terminus for constant or

decreasing ⟨f⟩ by step-size downshifting. It contrasts with skeletal actin and suggests their difference must lie in the different ELC N-terminus actin binding affinities implying the weaker affinity skeletal actin interaction distorts competition between lever arm strain-inhibited ADP release and ratcheting ELC N-terminus mechanisms. A constant or decreasing duty ratio may be a desirable characteristic for normal cardiac function by enforcing a constant or lower fraction of strongly bound myosins over most or all of the normal contraction cycle.

The notion that strain-inhibited ADP release is rate limiting for the myosin contraction cycle and that strain from the resisting load affects the ADP release to increase $t_{on}$ and ⟨f⟩ (equation (2.8)) presumes strain-inhibited ADP release solely regulates actomyosin cycling during muscle shortening. Measurements of $t_{on}$ for ensemble myosin indicate it is shorter by a factor of approximately 10 [33] when compared to estimates using myosin in a laser-trap [56], implying ADP release may not impact $t_{on}$. Data in figure 7*b* suggest the latter is true for cardiac actin when myosin is in its low-force regime. A notable alternative model suggests the rate of myosin attachment to actin limits contraction cycling [57].

## 3.4. Structure of the ELC/actin interaction in cardiac muscle

Figure 8 shows ventricular myosin bound to cardiac actin at the end of the powerstroke based on a skeletal protein structure in the equivalent configuration from Aydt *et al.* [18]. The figure shows the three actin monomers interacting with one myosin S1 and including the N-terminus ELC/actin interaction. The four cardiac actin residues differing from skeletal actin (Asp2, Glu3, Leu299 and Ser358) are depicted with space-filling atoms. Homology modelling indicates a potential hydrogen bond between side chains Ser358 O-H and Glu6 O on the βmys ELC N-terminus actin-binding domain. This interaction, and its homologue from all skeletal actomyosin, is shown in figure 9. The proposed hydrogen bond in the cardiac actomyosin is indicated with the blue dashed line and the H–O distance of 1.65 Å. In the skeletal actomyosin model, the closest approach of side chain heavy atoms from Thr358 (skeletal actin) and Val6 (skeletal ELC) is 3.9 Å.

Removing cardiac actin contaminant from tissue purified porcine cardiac myosin is well known to be less efficient than removing the skeletal actin contaminant from tissue purified rabbit skeletal myosin [4]. We hypothesize that the actin/ELC N-terminus contact for the cardiac actin and myosin isoforms stabilizes the cardiac actomyosin complex with the additional hydrogen bond between side chains Ser358 O-H and Glu6 O (figure 9) that is likely insensitive to ATP binding, thus frustrating efficient separation of cardiac actin from cardiac myosin in the presence of ATP.

## 4. Discussion

Skeletal and cardiac mammalian α-actin isoforms dominate in skeletal and cardiac muscle, respectively. They differ slightly from each other at four side chains where substitutions Glu2Asp, Asp3Glu, Leu299Met and Thr358Ser convert skeletal to cardiac actin. In normal adult humans, skeletal muscle is homogeneous in the skeletal α-actin [50] while cardiac muscle has cardiac : skeletal α-actin stoichiometry of approximately 4 : 1 [27]. Sequence similarity between isoforms promises

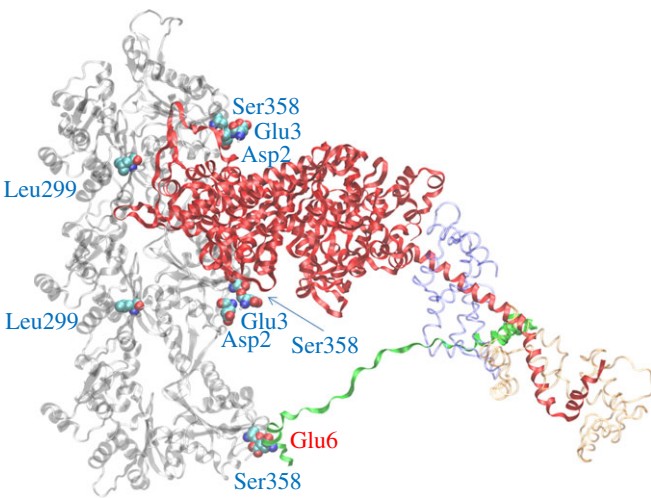

**Figure 8.** Homology modelled human ventricular myosin bound to cardiac actin making the ELC/actin contact at the end of the powerstroke. Three cardiac actin crystal structures in a filament in transparent grey have the top two actin monomers contacting with the myosin heavy chain (red) and the bottom actin monomer in contact with the myosin ELC N-terminus (green). Myosin RLC (transparent red) and ELC not including the N-terminus (transparent blue) are indicated. Residues in the cardiac actin sequence differing from the skeletal sequence are denoted by space-filling atoms with blue labels. Ser358 in the bottom actin monomer is proposed to form a hydrogen bond with one of the $\varepsilon$-oxygens in Glu6 (denoted by space-filling atoms with red label) from the ELC N-terminus (see figure 9).

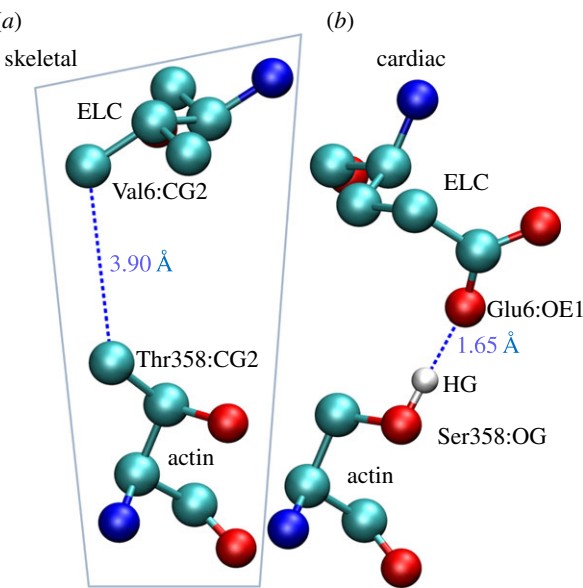

**Figure 9.** The actin Ser358 interaction with the ELC Glu6 from the cardiac isoforms (*b*) and the equivalent interaction for the skeletal actin and myosin sequences (*a* with boundary). The cardiac isoforms are proposed to enter into an O—H hydrogen bond depicted by the blue dashed line for atoms separated by 1.65 Å.

near identical three-dimensional structures; nevertheless, they are functionally distinguishable with relative upregulation of the skeletal version in the ventriculum accompanying hypertrophic cardiomyopathy (HCM) disease [28]. We compared βmys ensemble enzymatic, ensemble mechanical and single myosin mechanical characteristics when interacting with actin purified from skeletal and cardiac muscle. We find differences in the βmys ensemble performance due to the different actin substrates that is linked to a molecular mechanism using insights from Qdot assay single myosin mechanical characterization.

Actin-activated myosin ATPase $V_{max}$ and $K_m$ are statistically indistinguishable for the two actin isoforms, indicating these substrates do not significantly affect motor domain ATP free energy transduction. It suggests the skeletal/cardiac actin substitutions Glu2Asp and Asp3Glu negligibly impact myosin functionality. By contrast, ensemble βmys force–

velocity characteristics for skeletal and cardiac actin filaments in a Moore–Hill characterization [37,38] specialized for this application indicate significant differences in ensemble motility velocity and force–velocity that unequivocally demonstrate functional differences between the two actin substrates. The skeletal actin upregulates power suggesting that the dynamic heart tissue upregulates the skeletal actin substrate when heart disease compromises power production. We demonstrated using the Qdot assay that the molecular basis for the functional differences between by the two actin substrates lies in the strain-sensitive myosin regulation mechanisms represented in figure 2.

Single myosin mechanical characteristics are affected variously by the different actin isoforms. Unitary displacement classification constraints introduced here better separate signal from background, giving higher step-size resolution for distinguishing 3 from 4 nm step-sizes. Emergence of the short step-size at approximately 4 nm contributed by the solo-$3^+$ nm pathway (red in figure 2) marks the onset of the high-force regime for both actins. For cardiac actin, the high-force regime begins to impact myosin function when the *in vitro* muscle (one actin filament moving over myosin in the assay) encounters greater than or equal to 5 pN resisting force. Peak power for this isoform occurs for approximately 8 pN drag force. Skeletal actin, by contrast, reaches the high-force regime at greater than 0 load. Uneven onset of the high-force regime for skeletal versus cardiac actin by application of the resisting force implies dynamic hybrid force–velocity characteristics will emerge as the skeletal/cardiac actin fractional content increases in the disease-compromised cardiac muscle. The emergent muscle will tend to activate the solo-$3^+$ pathway at lower resisting force, causing an upswing in sliding velocity, duty ratio and power.

A more subtle distinction between the actin isoforms is characterized by the ensemble average duty ratio, ⟨f⟩. It shows the cardiac actin system uses actively competing mechanisms for lever arm strain-inhibited ADP release and the ELC ratchet to hold ⟨f⟩ constant over increasing load. It contrasts with skeletal actin and suggests their difference must lie in the different ELC N-terminus actin binding affinities, implying the weaker affinity skeletal actin interaction distorts competition between the lever arm strain mechanisms. A constant duty ratio over dynamic loading may be a desirable characteristic for

normal cardiac function by enforcing a constant fraction of strongly bound myosins over the normal contraction cycle in contrast to a modulating fraction presumably interfering with super-relaxation maintenance [58,59]. The novel force–velocity characteristics for βmys caused by the actin isoforms in the heart muscle, and that are progressively modulated with heart disease onset, are dependent mainly on the cardiac myosin ELC and its weaker interaction with the skeletal actin at Thr358. This insight, provided by the Qdot assay, elucidates the fundamental regulatory mechanisms in an autonomous myosin and identifies the site on actin responsible for the functional differences imposed by skeletal and cardiac actins.

In the absence of load, Qdot assay measured average and individual βmys step-sizes are 0.4–0.5 nm longer for the cardiac versus skeletal actin substrates. Contacts between βmys heavy chain and the actin isoforms are identical based on actin-activated myosin ATPase, hence it follows that differences between structures of the skeletal and cardiac actin rather than pathway flux regulation are responsible. Furthermore, residues differing between the isoforms do not fall within the contacts between G-actin monomers in F-actin [60,61], implying a global F-actin structural difference causes the longer unitary step-sizes for βmys bound to cardiac actin. The βmys lever-arm swing when myosin makes the powerstroke causes greater displacement of the cardiac versus skeletal actin, implying a greater nanometre per ATP efficiency under low loading for the heart. A minor efficiency improvement is a meaningful adaptation in a muscle that works constantly like the heart. Nonetheless, in conditions where energy conservation would also seem critical, such as in fatigue resistant skeletal muscle, βmys is present but not cardiac actin [62].

## 5. Conclusion

Ensemble βmys force–velocity characteristics demonstrate unequivocally a functional difference between skeletal and cardiac actin substrates. We investigated the molecular basis for the differences using the Qdot assay to compare single βmys step-size and step-frequency characteristics when moving loaded cardiac and skeletal actin. New Qdot assay data classification constraints introduced here better separate signal from background giving higher signal-to-noise data and permitting sub-nanometre step-size resolution. Using this new technology we show significant myosin step-size redistribution for skeletal versus cardiac actin substrate under loaded conditions. The skeletal and cardiac actin isoforms affect strain-sensitive regulation such that onset of the high-force regime, where an increased contribution from the solo-$3^+$ pathway contributes a new 4 nm step-size displacement, is offset to higher loads when myosin impels cardiac actin.

We attribute cardiac actin impact on myosin load sensitivity to the unique and specific cardiac ELC N-terminus/cardiac actin contact at Glu6/Ser358. It modifies the cardiac motor force–velocity characteristics by stabilizing the ELC N-terminus/cardiac actin association. Higher affinity at this site frustrates efficient actomyosin dissociation by ATP during βmys purification leading to more actin contamination (figure 1), and is observed functionally to enhance long (8 nm) and short (3 nm) step-size probabilities at the expense of the intermediate (5 nm) step-size probability in the low-force regime where loads are less than 5 pN. These observations combined with the observation of a constant ensemble average duty ratio, ⟨f⟩, show that the cardiac myosin/cardiac actin system uses actively competing mechanisms of lever arm strain-inhibited ADP release and the ELC ratchet for maintaining approximately constant ⟨f⟩ over increasing load by step-size downshifting. By contrast, the skeletal actin substrate in the cardiac myosin/skeletal actin system drives it into the high-force regime at lower loads to upregulate the solo-$3^+$ nm pathway flux, power and ⟨f⟩. The latter characteristics are progressively upmodulated with HCM disease onset.

The three-dimensional cardiac F-actin structure was shown to be modified compared to skeletal F-actin conformation. The structural change imparts a subtle but significant 0.4–0.5 nm myosin displacement advantage to the cardiac F-actin *in vitro*. It implies a greater efficiency for actin displacement versus ATP consumed.

Skeletal and cardiac actins induce unique cardiac myosin force–velocity characteristics to modulate autonomous myosin function in the heart. Qdot assay experiments implicate the cardiac actin/myosin ELC N-terminus contact at S358/E6 as the structural origin of the myosin modulatory mechanism. Cardiac-specific expression of cardiac actin and the ELC N-terminus, and their possible joint role in regulating myosin motor function in the heart, suggests a target for re-programming autonomous myosin function in a poorly functioning heart.

Ethics. Protocols were approved by the Institutional Animal Care and Use Committee at Mayo Clinic Rochester (protocol A56513-13-R16).

Data accessibility. Data used in this study are provided in summary form in the text. Representative raw Qdot assay data are deposited at Zenodo (search 86709883e2fc643bf67a61c84e66b2e0).

Authors' contributions. Y.W. conceived and designed experiments, performed experiments and analysed data. K.A. conceived and designed experiments, and contributed actins. T.P.B. conceived and designed experiments, analysed data and wrote the paper. All authors gave final approval for publication.

Competing interests. We have no competing interests.

Funding. This work was supported by the Mayo Foundation.

Acknowledgement. We thank Susanna P. Garamszegi for excellent technical assistance with preparation and characterization of the cardiac actin.

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
