## [Reviewer comments · Open Biology]

Review History

RSOB-18-0143.R0 (Original submission)

Review form: Reviewer 1

Recommendation

Accept with minor revision (please list in comments)

Are each of the following suitable for general readers?

- a) **Title**
Yes
- b) **Summary**
Yes
- c) **Introduction**
Yes

Is the length of the paper justified?

Yes

Should the paper be seen by a specialist statistical reviewer?

No

Is it clear how to make all supporting data available?

Yes

Is the supplementary material necessary; and if so is it adequate and clear?

Not Applicable

Do you have any ethical concerns with this paper?

No

Comments to the Author

Wang and his co-authors investigated important aspects of the regulatory mechanisms between actin and myosin in cardiac cells. In the past limited attention has been paid to the roles of actin isoforms, this paper provides several interesting lines of those functional details. The manuscript presents data and considerations from experiments where the enzymatic, mechanical and single molecule mechanistic properties of the interaction between cardiac ventricular myosin and cardiac or skeletal actin isoforms were investigated. The Introduction chapter properly summarizes the information available in the field. The experimental procedures and the data analysis is properly described, the new Qdot assay is attractive and gives a special importance to the manuscript. The authors showed that while some of the basic enzymatic parameters were not affected by the change in the actin isoform content, skeletal actin supported larger isometric force than cardiac actin. The conclusions the authors present are based on their observations, and they are also interesting and important. They provide a simple model where the interaction of cardiac ventricular myosin with different actin isoforms have a regulatory role in healthy and also in pathological conditions.

I suggest this manuscript to be accepted as it is, or after the authors reflected to my minor comments below. I think in the latter case the paper could be more understandable for the general readers.

Comments, questions:

It is known that in normal mammalian adult heart muscle the ratio of skeletal to cardiac actin isoforms is appr. 1:4. It would probably be interesting to see in experiments similar to those in those manuscript how the mixture of actin isoforms would interact with myosin. Would they provide results between those observed for the two isoforms separately?

How should the reader imagine the mixture of cardiac and skeletal actin isoforms? Would each of the filaments contain the corresponding cardiac and skeletal protomers in a homogeneous distribution along the filaments? Or would they form separate filaments after polymerisation?

Considering the observed differences between the interactions of cardiac ventricular myosin with skeletal or cardiac actin; what are the myosin ATPase kinetic steps which possibly altered by the replacement of the actin isoform?

In the experiments with phalloidin, could this drug have any effect on the actin isoform dependence? One can imagine that the effect of phalloidin on the dynamic properties of actin

may alter its force supporting function two, that can then be isoform dependent.

Review form: Reviewer 2

Recommendation

Accept with minor revision (please list in comments)

Are each of the following suitable for general readers?

- a) **Title**
Yes
- b) **Summary**
Yes
- c) **Introduction**
Yes

Is the length of the paper justified?

Yes

Should the paper be seen by a specialist statistical reviewer?

No

Is it clear how to make all supporting data available?

Yes

Is the supplementary material necessary; and if so is it adequate and clear?

Not Applicable

Do you have any ethical concerns with this paper?

No

Comments to the Author

The manuscript "Cardiac and Skeletal Actin Substrates Uniquely Tune Cardiac Myosin Strain-Dependent

Mechanics" by Wang et al. examines the force-velocity relationship for β -MHC (β mys) propelling skeletal vs. cardiac actin using ensemble in vitro motility assays and single myosin Q-dot assays. Cardiac and skeletal actins have >98% sequence identity and only differ at positions 2 and 3 where Asp-Glu reverses to Glu-Asp, at Met299Leu and at Ser358Thr for cardiac/skeletal actins. Using their improved Q-dot assay, with a higher signal-to-noise ratio, the authors generated sets of data interrogating single β mys step-size and step-frequency characteristics when moving cardiac vs. skeletal actin under loaded conditions. While no differences are seen in actin-activated myosin ATPase activity, they observed significant differences in myosin step-size distribution between skeletal and cardiac actin under load related to strain-dependent mechanics. The differences originate, at least in part, from the way the cardiac vs. skeletal actin isoforms interact with β mys and the interaction between the C-terminus of actin and the N-ELC terminus of myosin (Fig. 9: stronger interaction for cardiac vs. skeletal actin). The structural change in actin conformation imparts a subtle but significant 0.4-0.5 nm myosin displacement advantage to the cardiac actin in vitro. It suggests a greater efficiency for actin displacement vs. ATP consumed.

This is a very important study providing insights into how subtle differences in the amino acid sequence between the cardiac and fast-twitch skeletal actin isoforms lead to substantial differences in the actin-myosin interaction when examined under load (power). Results from this paper will be extremely important to a wide research community performing standard biochemical/biophysical assays of actin – myosin interaction in vitro and especially those using actin in characterizing the effect of cardiomyopathy-causing mutations in cardiac myosin on β mys function. It also provides an explanation as to why skeletal muscle actin is often upregulated in cardiac disease. The skeletal/cardiac actin stoichiometry is ~1:4 in a normal adult human heart and the upregulation of skeletal isoform is observed in diseased myocardium. Fig. 4 c shows that skeletal actin upregulates power compared with cardiac and suggests that the dynamic heart tissue upregulates the skeletal actin substrate when heart disease compromises power production.

In conclusion, the findings of the manuscript are significant, novel and form an unquestioned strength. It is anticipated that the study will be of interest to the broad readership of Open Biology and therefore it is highly recommended for publication.

Specific comments to authors:

1. Introduction: Fig. 8 could be introduced as Fig. 1 in the text of the Introduction where the authors elaborate on differences between the skeletal and cardiac actin isoforms (p. 6, lines 91-101). The overlapped actin sequences (cardiac vs. skeletal) could be supported by the ELC amino-acid sequence with highlighted residues important for the actin-N-ELC binding.
2. Methods are very elaborate (p.7-20) and could be simplified by (1) referencing the authors' previous publications and leaving only those parts describing new developments that are relevant to the current study; or (2) moved, at least in part, to the supplement.
3. The reason for bringing Fig. 2 (4-pathway contraction model) from the previously published paper (ref. 25) into the Methods of the current manuscript is not explained or justified.
4. Results: Fig. 4 is of great importance to the understanding of physiological differences between cardiac vs. skeletal actin, and the presentation of the figure could be improved. The middle panel should have x-axes shown clearly just as top and bottom panels. Presentation of panels d, e, and f needs more explanation than just "Figure 4 panels d–f parallel those in panels a–c except that [α] sampling is limited to 0, 1, 3, and 5 μ g/ml".
5. It would be beneficial to the readers to have each figure concluded on after providing methodological details (Figs. 5, 6). The explanation is often provided later in the Results section rather than after the first mention of the figure.
6. Discussion often sounds like a continuation of Results (e.g. p.28/29) rather than providing an explanation/interpretation.
7. Conclusion: The authors should avoid referencing to specific figures here and provide "a take home message(s)" from their studies.

Decision letter (RSOB-18-0143.R0)

17-Oct-2018

Dear Dr Burghardt

We are pleased to inform you that your manuscript RSOB-18-0143 entitled "Cardiac and Skeletal Actin Substrates Uniquely Tune Cardiac Myosin Strain-Dependent Mechanics" has been accepted

by the Editor for publication in Open Biology. The reviewer(s) have recommended publication, but also suggest some minor revisions to your manuscript. Therefore, we invite you to respond to the reviewer(s)' comments and revise your manuscript.

Please submit the revised version of your manuscript within 14 days. If you do not think you will be able to meet this date please let us know immediately and we can extend this deadline for you.

- 1) A text file of the manuscript (doc, txt, rtf or tex), including the references, tables (including captions) and figure captions. Please remove any tracked changes from the text before submission. PDF files are not an accepted format for the "Main Document".
- 2) A separate electronic file of each figure (tiff, EPS or print-quality PDF preferred). The format should be produced directly from original creation package, or original software format. Please note that PowerPoint files are not accepted.
- 3) Electronic supplementary material: this should be contained in a separate file from the main text and meet our ESM criteria (see <http://royalsocietypublishing.org/instructions-authors#question5>). All supplementary materials accompanying an accepted article will be treated as in their final form. They will be published alongside the paper on the journal website and posted on the online figshare repository. Files on figshare will be made available approximately one week before the accompanying article so that the supplementary material can be attributed a unique DOI.

Online supplementary material will also carry the title and description provided during submission, so please ensure these are accurate and informative. Note that the Royal Society will not edit or typeset supplementary material and it will be hosted as provided. Please ensure that the supplementary material includes the paper details (authors, title, journal name, article DOI). Your article DOI will be 10.1098/rsob.2016[last 4 digits of e.g. 10.1098/rsob.20160049].

- 4) A media summary: a short non-technical summary (up to 100 words) of the key findings/importance of your manuscript. Please try to write in simple English, avoid jargon, explain the importance of the topic, outline the main implications and describe why this topic is newsworthy.

Images

Data-Sharing

It is a condition of publication that data supporting your paper are made available. Data should be made available either in the electronic supplementary material or through an appropriate repository. Details of how to access data should be included in your paper. Please see <http://royalsocietypublishing.org/site/authors/policy.xhtml#question6> for more details.

Data accessibility section

Sincerely,

The Open Biology Team
<mailto:openbiology@royalsociety.org>

ditage Insights by clicking on the following link: <https://www.surveymonkey.com/r/author-perspectives-on-academic-publishing-royal-society>

This should take no more than 15 minutes and you will have the opportunity to enter a prize draw. We hope these results will provide us with valuable insights we can use to improve our service.

Reviewer(s)' Comments to Author:

Referee: 1

Comments to the Author(s)

Wang and his co-authors investigated important aspects of the regulatory mechanisms between actin and myosin in cardiac cells. In the past limited attention has been paid to the roles of actin isoforms, this paper provides several interesting lines of those functional details. The manuscript presents data and considerations from experiments where the enzymatic, mechanical and single molecule mechanistic properties of the interaction between cardiac ventricular myosin and cardiac or skeletal actin isoforms were investigated. The Introduction chapter properly summarizes the information available in the field. The experimental procedures and the data analysis is properly described, the new Qdot assay is attractive and gives a special importance to the manuscript. The authors showed that while some of the basic enzymatic parameters were not affected by the change in the actin isoform content, skeletal actin supported larger isometric force than cardiac actin. The conclusions the authors present are based on their observations, and they are also interesting and important. They provide a simple model where the interaction of cardiac ventricular myosin with different actin isoforms have a regulatory role in healthy and also in pathological conditions.

I suggest this manuscript to be accepted as it is, or after the authors reflected to my minor comments below. I think in the latter case the paper could be more understandable for the general readers.

Comments, questions:

It is known that in normal mammalian adult heart muscle the ratio of skeletal to cardiac actin isoforms is appr. 1:4. It would probably be interesting to see in experiments similar to those in those manuscript how the mixture of actin isoforms would interact with myosin. Would they provide results between those observed for the two isoforms separately?

How should the reader imagine the mixture of cardiac and skeletal actin isoforms? Would each of the filaments contain the corresponding cardiac and skeletal protomers in a homogeneous distribution along the filaments? Or would they form separate filaments after polymerisation?

Considering the observe differences between the interactions of cardiac ventricular myosin with skeletal or cardiac actin; what are the myosin ATPase kinetic steps which possibly altered by the replacement of the actin isoform?

In the experiments with phalloidin, could this drug have any effect on the actin isoform dependence? One can imagine that the effect of phalloidin on the dynamic properties of actin may alter its force supporting function two, that can then be isoform dependent.

Referee: 2

Comments to the Author(s)

The manuscript "Cardiac and Skeletal Actin Substrates Uniquely Tune Cardiac Myosin Strain-Dependent

Mechanics" by Wang et al. examines the force-velocity relationship for β -MHC (β mys) propelling skeletal vs. cardiac actin using ensemble in vitro motility assays and single myosin Q-dot assays. Cardiac and skeletal actins have \approx 98% sequence identity and only differ at positions 2 and 3 where Asp-Glu reverses to Glu-Asp, at Met299Leu and at Ser358Thr for cardiac/skeletal actins. Using their improved Q-dot assay, with a higher signal-to-noise ratio, the authors generated sets of data interrogating single β mys step-size and step-frequency characteristics when moving cardiac vs. skeletal actin under loaded conditions. While no differences are seen in actin-activated myosin ATPase activity, they observed significant differences in myosin step-size distribution between skeletal and cardiac actin under load related to strain-dependent mechanics. The differences originate, at least in part, from the way the cardiac vs. skeletal actin isoforms interact with β mys and the interaction between the C-terminus of actin and the N-ELC terminus of myosin (Fig. 9: stronger interaction for cardiac vs. skeletal actin). The structural change in actin conformation imparts a subtle but significant 0.4-0.5 nm myosin displacement advantage to the cardiac actin in vitro. It suggests a greater efficiency for actin displacement vs. ATP consumed.

This is a very important study providing insights into how subtle differences in the amino acid sequence between the cardiac and fast-twitch skeletal actin isoforms lead to substantial differences in the actin-myosin interaction when examined under load (power). Results from this paper will be extremely important to a wide research community performing standard biochemical/biophysical assays of actin - myosin interaction in vitro and especially those using actin in characterizing the effect of cardiomyopathy-causing mutations in cardiac myosin on β mys function. It also provides an explanation as to why skeletal muscle actin is often upregulated in cardiac disease. The skeletal/cardiac actin stoichiometry is \sim 1:4 in a normal adult human heart and the upregulation of skeletal isoform is observed in diseased myocardium. Fig. 4

c shows that skeletal actin upregulates power compared with cardiac and suggests that the dynamic heart tissue upregulates the skeletal actin substrate when heart disease compromises power production.

In conclusion, the findings of the manuscript are significant, novel and form an unquestioned strength. It is anticipated that the study will be of interest to the broad readership of Open Biology and therefore it is highly recommended for publication.

Specific comments to authors:

1. Introduction: Fig. 8 could be introduced as Fig. 1 in the text of the Introduction where the authors elaborate on differences between the skeletal and cardiac actin isoforms (p. 6, lines 91-101). The overlapped actin sequences (cardiac vs. skeletal) could be supported by the ELC amino-acid sequence with highlighted residues important for the actin-N-ELC binding.
2. Methods are very elaborate (p.7-20) and could be simplified by (1) referencing the authors' previous publications and leaving only those parts describing new developments that are relevant to the current study; or (2) moved, at least in part, to the supplement.
3. The reason for bringing Fig. 2 (4-pathway contraction model) from the previously published paper (ref. 25) into the Methods of the current manuscript is not explained or justified.
4. Results: Fig. 4 is of great importance to the understanding of physiological differences between cardiac vs. skeletal actin, and the presentation of the figure could be improved. The middle panel should have x-axes shown clearly just as top and bottom panels. Presentation of panels d, e, and f needs more explanation than just "Figure 4 panels d-f parallel those in panels a-c except that [a] sampling is limited to 0, 1, 3, and 5 $\mu\text{g}/\text{ml}$ ".
5. It would be beneficial to the readers to have each figure concluded on after providing methodological details (Figs. 5, 6). The explanation is often provided later in the Results section rather than after the first mention of the figure.
6. Discussion often sounds like a continuation of Results (e.g. p.28/29) rather than providing an explanation/interpretation.
7. Conclusion: The authors should avoid referencing to specific figures here and provide "a take home message(s)" from their studies.

Author's Response to Decision Letter for (RSOB-18-0143.R0)

See Appendix A.

Decision letter (RSOB-18-0143.R1)

26-Oct-2018

Dear Dr Burghardt

We are pleased to inform you that your manuscript entitled "Cardiac and Skeletal Actin Substrates Uniquely Tune Cardiac Myosin Strain-Dependent Mechanics" has been accepted by the Editor for publication in Open Biology.

You can expect to receive a proof of your article from our Production office in due course, please

check your spam filter if you do not receive it within the next 10 working days. Please let us know if you are likely to be away from e-mail contact during this time.

Article processing charge

Please note that the article processing charge is immediately payable. A separate email will be sent out shortly to confirm the charge due. The preferred payment method is by credit card; however, other payment options are available.

Sincerely,

The Open Biology Team
mailto:openbiology@royalsociety.org

Appendix A

Response to Reviewers

RE: Open Biology - Manuscript RSOB-18-0143

We value the detailed commentary made by the referees and have revised the manuscript to address their concerns as described below.

Referee 1

1. It is known that in normal mammalian adult heart muscle the ratio of skeletal to cardiac actin isoforms is appr. 1:4. It would probably be interesting to see in experiments similar to those in those manuscript how the mixture of actin isoforms would interact with myosin. Would they provide results between those observed for the two isoforms separately?

The comment is highly pertinent to investigating cardiac actin function in the heart. We are cognizant of the 1:4 stoichiometry in normal adult human (and bovine) ventriculum and discussed possible effects of the actin mixture on our data (pg 18, starting line 326). There are other complications to consider as noted in the referee's next comment. We can address these issues in future work by varying skeletal actin content in the mixture and by preparation of a pure cardiac actin sample.

2. How should the reader imagine the mixture of cardiac and skeletal actin isoforms? Would each of the filaments contain the corresponding cardiac and skeletal protomers in a homogeneous distribution along the filaments? Or would they form separate filaments after polymerisation?

This interesting issue has not been investigated to our knowledge. As mentioned above, we will probably need to prepare the pure cardiac actin isoform for new experiments on cardiac actin polymerization and cardiac myosin motility.

3. Considering the observe differences between the interactions of cardiac ventricular myosin with skeletal or cardiac actin; what are the myosin ATPase kinetic steps which possibly altered by the replacement of the actin isoform?

Our results show myosin functionality differs for the skeletal vs cardiac actin substrates when the system is under load. Strain dependent steps (see **Figure 2**) seem most likely to be implicated in these conditions. Our **Figure 6** shows

a strongly down-regulated 5 nm step-size and up-regulated 8 nm step-size for the cardiac compared to skeletal actin with the imposition of light resisting force. Myosin dynamics change at higher force as the cardiac actin muscle shifts into the high force regime. Multiple parallel mechanical pathways make it tricky to identify the rate affected in the high force regime. Certainly lower ADP release rate down-regulates the 5 nm step-size in the cardiac compared to skeletal actin at intermediate force (1-2 pN).

4. In the experiments with phalloidin, could this drug have any effect on the actin isoform dependence? One can imagine that the effect of phalloidin on the dynamic properties of actin may alter its force supporting function two, that can then be isoform dependent.

This is an interesting hypothesis with consequences for all in vitro motility assays. It was shown that phalloidin and jasplakinolide promote actin filament stability that is propagated cooperatively over the filament [1]. Either or both drugs might alter actin filament compliance and consequently myosin strain sensitive behavior. We use biotin-phalloidin to stabilize actin and to link the streptavidin-Qdot to the actin filament in the motility assay. An alternative actin stabilizing drug would provide contrast to the phalloidin-actin system elucidating the effects on myosin strain sensitive behavior.

Referee #2

1. Introduction: Fig. 8 could be introduced as Fig. 1 in the text of the Introduction where the authors elaborate on differences between the skeletal and cardiac actin isoforms (p. 6, lines 91-101). The overlapped actin sequences (cardiac vs. skeletal) could be supported by the ELC amino-acid sequence with highlighted residues important for the actin-N-ELC binding.

We disagree with this thoughtful suggestion because **Figure 8** is based on the homology modeling of cardiac actomyosin using a skeletal actomyosin structure template [2]. As such, it needs methods and background context to be comprehensible to the reader and reproducible by others. The context information is beyond the scope of the Introduction.

2. Methods are very elaborate (p.7-20) and could be simplified by (1) referencing the authors' previous publications and leaving only those parts describing new developments that are relevant to the current study; or (2) moved, at least in part, to the supplement.

We agree and have made adjustments to the Methods. The section *Qdot Assay Event-Velocity Histogram Simulation* (pg 13, starting line 221) has been mostly removed in favor of referencing previous work. We retained only the portion describing the new baseline subtraction method.

We retained the section: *Force calibration in the loaded actin in vitro motility and Qdot assays* (pg 11, starting line 187) where we describe a new method involving dimensionless parameters K (proportional to myosin attachment rate f_{APP}) and normalized force in the force-velocity relationship.

Other sections retained contain brief descriptions of protein preparation protocols, unique statistical information, or methods we have not used previously.

3. *The reason for bringing Fig. 2 (4-pathway contraction model) from the previously published paper (ref. 25) into the Methods of the current manuscript is not explained or justified.*

The model here is modestly changed from previous work. In **Figure 2**, the slip transition in the solo 3^+ nm pathway is more accurately defined. We justify inclusion of **Figure 2** in the section *4-pathway contraction model* in the revised manuscript (pg 15, lines 258-265).

4. *Results: Fig. 4 is of great importance to the understanding of physiological differences between cardiac vs. skeletal actin, and the presentation of the figure could be improved. The middle panel should have x-axes shown clearly just as top and bottom panels. Presentation of panels d, e, and f needs more explanation than just “Figure 4 panels d–f parallel those in panels a–c except that $[\alpha]$ sampling is limited to 0, 1, 3, and 5 $\mu\text{g/ml}$ ”.*

We agree and have made the suggested revision to Figure 4 middle panels. Our result summary for Figure 4 panels d-f covers a good deal more detail than suggested in the comment. Please see pg 22 lines 411-426 in the revision.

5. *It would be beneficial to the readers to have each figure concluded on after providing methodological details (Figs. 5, 6). The explanation is often provided later in the Results section rather than after the first mention of the figure.*

We disagree with this comment. A small delay in summarizing findings from **Figures 5 & 6** allows their integration with the ensemble averaged quantities

in **Figure 7**. We believe this strategy is more coherent. We have also edited this section slightly for clarity by directing the reader to the original literature for computing <d> and <f>. This is important in the revised manuscript because we removed pertinent equations from the Methods section in response to comment 2 above.

6. Discussion often sounds like a continuation of Results (e.g. p.28/29) rather than providing an explanation/interpretation.

We revised Discussion in response to this comment by emphasizing interpretation over reviewing results. See Discussion starting pg 31.

7. Conclusion: The authors should avoid referencing to specific figures here and provide “a take home message(s)” from their studies.

We thoroughly revised Conclusion in response to this comment. See Conclusion starting pg 34.

References

- 1 Visegrady, B., Lorinczy, D., Hild, G., Somogyi, B., Nyitrai, M. 2005 A simple model for the cooperative stabilisation of actin filaments by phalloidin and jasplakinolide. *FEBS Lett.* **579**, 6-10.
- 2 Ayd, E. M., Wolff, G., Morano, I. 2007 Molecular modeling of the myosin-S1(A1) isoform. *J. Struct. Biol.* **159**, 158-163.
(<https://doi.org/10.1016/j.jsb.2007.04.002>)